# Agri-Food By-Products in Cancer: New Targets and Strategies

**DOI:** 10.3390/cancers14225517

**Published:** 2022-11-10

**Authors:** Carmela Sorrentino, Martina Di Gisi, Giulia Gentile, Fabrizio Licitra, Rosa D’Angiolo, Pia Giovannelli, Antimo Migliaccio, Gabriella Castoria, Marzia Di Donato

**Affiliations:** Department of Precision Medicine, University of Campania “L. Vanvitelli”, 80138 Naples, Italy

**Keywords:** agri-food by-products, phenolic compounds, breast cancer, prostate cancer, gastrointestinal cancer, EGFR

## Abstract

**Simple Summary:**

Bio-sustainability is one of the more attractive challenges of this era. About one-third of the food placed on the market is wasted, bringing economic and environmental implications. This review focuses on the role of natural derivatives from agri-food by-products to encourage the processes of the circular economy. Activities related to the reuse of agricultural processing waste could contribute to the birth of innovative companies and start-ups able to work for environmental sustainability. Agri-food by-products contain bioactive compounds that could be employed in the production of enriched food, cosmetics, and drugs by pharmaceutics companies. Furthermore, this manuscript aims to assess the main targets of these natural derivatives with particular attention on Epidermal growth factor receptor in breast, prostate and colorectal cancers.

**Abstract:**

The globalization and the changes in consumer lifestyles are forcing us to face a deep transformation in food demand and in the organization of the entire food production system. In this new era, the food-loss and food-waste security nexus is relevant in the global debate and avoiding unsustainable waste in agri-food systems as well as the supply chain is a big challenge. “Food waste” is useful for the recovery of its valuable components, thus it can assume the connotation of a “food by-product”. Sustainable utilization of agri-food waste by-products provides a great opportunity. Increasing evidence shows that agri-food by-products are a source of different bioactive molecules that lower the inflammatory state and, hence, the aggressiveness of several proliferative diseases. This review aims to summarize the effects of agri-food by-products derivatives, already recognized as promising therapeutics in human diseases, including different cancer types, such as breast, prostate, and colorectal cancer. Here, we examine products modulating or interfering in the signaling mediated by the epidermal growth factor receptor.

## 1. Introduction

The World Cancer Research Fund (WCRF) states that correct eating habits have an important role in the prevention of human diseases and cancer [1]. Vegetables and fruits or their components seem to play a healthy role. In addition to genetic variations and other factors, a low consumption of vegetables, wholegrains, legumes, and spices is involved in the development and progression of cancer and other chronic diseases. Consistent with the geographic area, the incidence rate of these diseases is different and seems directly linked to diet and the available resources. In the south of Europe and in Southeast Asia, the incidence of developing cancer is lower compared with the northern European population or populations living in more industrialized areas [2]. Thus, dietary optimization is crucial for cancer prevention and almost 20% of cancers are preventable with a diet containing a high amount and variety of vegetables and fruits [3].

In 2010, the Food and Agriculture Organization (FAO) defined “sustainable diets” the “diets with low environmental impacts, which contribute to food and nutrition security and to healthy life for present and future generations” [4]. Currently, about one third of the food placed on the market is wasted since it is considered not useful for the supply. The definition of “food waste” acquires a negative meaning, not only because of the economic and environmental implications, but also because these products might have potential and could be useful for the recovery of valuable components, thus assuming the connotation of “food by-products” [5]. These products could become part of the circular economy, recovering the percentage of by-products lost [6]. In particular, agri-food by-products have been increasingly considered for the extraction of new nutraceuticals, functional foods, and food additives, which could increase the quantity and quality of natural bioactive compounds available and contribute to the development of sustainability [7]. Peel, seeds, shells, pomace, and leaves contain bioactive compounds (e.g., phenols, antioxidants, anthocyanins, carotenoids, fatty acids, and peptides) as well as fibers and enzymes employed in the production of enriched food, cosmetics, and drugs by pharmaceutics companies [8,9]. Great attention is given to polyphenols or phenolic compounds (PhCs) that have a preventive role in inflammation as well as chronic diseases related to oxidative stress, because of their scavenger activity [10]. Dietary polyphenols constitute one of the most numerous groups of natural products. They are all characterized by a common chemical structure comprising an aromatic ring with one or more hydroxyl substituents, but the natural products belonging to this category are very heterogeneous and can be divided into several classes. The main groups include flavonoids, phenolic acids, lignans, chacones, stilbenes, and anthocianins (Figure 1). Variations in the position and nature of the substituents in the aromatic hydrocarbon backbones lead to a variety of molecules.

These natural substances prevent aging [10], gastric, duodenal [11], cardiovascular [12] and neurodegenerative [13] diseases, type 2 diabetes [14], and certain types of cancer [15,16,17,18]. They are particularly abundant in citrus fruits, such as oranges (*Citrus sinensis*), lemons (*Citrus limon*), limes (*Citrus aurantifolia*), grapefruit (*Citrus paradise*), but also grapes (*Vitis vinifera* L.), annurca apples (*annurca* or *Malus domestica*), and olives (*Olea Europea Sativa*) [19,20,21,22,23,24]. Evidence shows that PhCs can interfere in the onset and progression of human diseases and cancer by affecting the growth factor (GF) and growth factor receptor (GFR) signaling circuits [25]. In normal cells, GFs mediate proliferation, differentiation, and migration required for normal tissue growth, organization, and repair. Derangements of GF signaling or availability, together with mutations or overexpression of their cognate receptors are often responsible for uncontrolled cell proliferation and differentiation leading to carcinogenesis [26]. Considerable attention has been drawn to the link occurring between the epidermal growth factor receptor (EGFR) and agri-food by-products. The EGFR has been extensively studied for the development of anticancer therapies [27]. In addition, the EGFR plays a crucial role in several inflammatory conditions, such as thrombin-mediated inflammation [28], virus-induced respiratory inflammations [29], palmitic acid-induced inflammation of cardiac muscle cells [30], neuroinflammation [31,32], inflammation related to skin diseases [33], and several cancer types, such as liver, gastrointestinal [34], breast [35], lung [36], and prostate [37] cancers.

Inflammation is a very important aspect of cancer, since many promoters, such as chemical irritants, partial organ resection, chronic stimulation by hormones might induce proliferation or recruit inflammatory cells, thus increasing the production of reactive oxygen species (ROS), which results in oxidative DNA damage, reduced DNA repair, and induction of a chronic inflammatory state [38]. Thus, inflammatory cells surrounding the tumor mass, favor the tumor development and produce a favorable environment for tumor growth, easing genomic instability and angiogenesis. In this context, anti-inflammatory agents would be efficacious towards early neoplastic progression and malignant conversion. Increasing evidence shows that dietary PhCs interfere not only with the EGFR-mediated signaling but also with the aggressiveness and the inflammatory state of the tumor. PhCs might also act as cytotoxic agents when used at high concentrations in cancer cells.

The present review aims to address the dual role of PhCs and summarize the effects of agri-food by-products derivatives, which are largely recognized as promising therapeutics in human diseases and cancer. Focus will be given to products affecting the EGFR-mediated signaling.

## 2. Agri-Food By-Products in Human Diseases

ROS originate from many endogenous or exogenous sources, including nutrients, growth factors, microbiome, cytokines, radiation, and other metabolisms. Oxidative stress derived from an excessive ROS production promotes the damage of DNA, RNA, and proteins, and may speed-up the normal aging process. It is involved in the pathogenesis of different chronic diseases, including atherosclerosis, diabetes, neurological disorders, cardiovascular disease, immune dysfunction, and cancer [38,39,40]. Oxidative-mediated reactions interfere with many fundamental processes, such as mitochondrial respiration, lipid synthesis, metal ion metabolism, phagocytosis, and xenobiotic biotransformation of organic compounds. Some antioxidant compounds or specific enzymes can neutralize the ROS to protect the integrity of cells or tissues. Sometimes, the delicate balance between antioxidants and ROS might be affected by exposure to toxic agents, leading to oxidative and inflammatory damage to DNA, proteins, or lipids [41,42,43,44]. This imbalance promotes chronic and age-related diseases. Currently, the fine balance of the ROS dose–cellular response curve is widely discussed. As already described, excessive ROS production results in cell necrosis and apoptosis. In contrast, subtoxic ROS levels activate signaling pathways and modulate gene transcription leading to cellular proliferation. It is still unclear whether antioxidants are more able to act as scavengers in the presence of low levels of ROS, or if there is a linear curve of dose-dependence. In addition to investigating the impact of ROS, it is important to highlight the beneficial effects of antioxidants.

Agri-food by-products have received growing attention for their role as preventive agents against oxidative damage, and good allies in relieving inflammatory states [45]. Of relevance is the role of PhCs as antioxidants and antiaging, anti-inflammatory, and antiproliferative agents. Among them, flavonoids, phenolic acids, and tannins modulate carbohydrate and lipid metabolism, reduce hyperglycemia and dyslipidemia, interfere with insulin resistance, improve pancreatic β-cell function and adipose tissue metabolism, stimulate insulin secretion, and alleviate oxidative stress, stress-sensitive signaling pathways, and inflammatory processes [46,47,48,49,50]. PhCs might prevent long-term diabetes’ complications, including neuropathy, microangiopathies, and macroangiopathies. They also improve fitness by decreasing the risk of metabolic syndrome [49] and protect against ROS-induced cell apoptosis in neurodegenerative disorders [51]. Indeed, they counteract the oxidative stress responsible for the activation of different transcription circuits (e.g., those exerted by nuclear factor erythroid 2–related factor 2 (Nrf2), nuclear factor κB (NF-κB), activator protein-1 (AP-1), peroxisome proliferator-activated receptor-γ (PPAR-γ), p53, hypoxia-inducible factor-1α (HIF-1α), and Wnt/β-catenin system), which mediate the expression of over 500 genes coding for GFs and their cognate receptors, cell cycle regulators, and inflammatory cytokines, resulting in chronic inflammation, cancer, diabetes, neurodegeneration, cardiovascular, and pulmonary diseases [52]. Table 1 summarizes the principal agri-food by-products with antioxidant capability and the relative diseases in which they are effective.

## 3. Agri-Food By-Products in Cancer: Antioxidant or Cytotoxic Agents?

Different epidemiological, preclinical, and early clinical studies suggest that chemicals derived from natural products may play a role in cancer prevention and treatment. The role in cancer prevention might be due to their ability to inhibit oxidative stress. Chronic inflammation, indeed, is a predisposing factor to different forms of cancer [61]. Cancer cells closely and continuously interact with the tumor microenvironment (TME), influencing it through the release of soluble signals. The TME is made up of different types of cells, such as myofibroblasts, endothelial, and inflammatory cells [62] that form a real inflammatory TME linked to cancer initiation, promotion, and progression. Hallmarks of inflammation in cancer include the infiltration of white blood cells, the presence of cytokines and chemokines, and the occurrence of tissue remodeling and angiogenesis [63]. Notably, the relationship between fibroblasts and ROS is undeniable. On one hand, ROS drive the transformation of fibroblasts into proinvasive and activated myofibroblasts through the upregulation of hypoxia-inducible factor (HIF1α) [64]; on the other, cancer-associated fibroblasts (CAFs) release high levels of H_2_O_2_ inducing carcinogenesis [65]. Use of nonsteroidal anti-inflammatory drugs (NSAIDs) to reduce cancer risk is, however, still debated. Many findings suggest that NSAIDs exhibit inhibitory effects in the pathogenesis of carcinogenesis [66,67,68]. Long-term aspirin use is associated to a reduction in the incidence and mortality for several cancer types, particularly with adenocarcinoma of esophagus, colorectal and stomach cancers. In addition, some effects have been also observed in patients affected by breast, lung, and prostate cancers [66]. Although NSAIDs seem to have beneficial effects in some cancer types, their prolonged use often induces side-effects. The use of natural substances, such as compounds derived from agri-food by-products, then emerges. Flavonoids, PhCs, carotenoids, and tocopherols are able to scavenge the free radicals, and act as reductants [69]. In contrast, agri-food by-products derivatives can also be evaluated for cancer treatment. Although most of the beneficial effects of natural compounds can be ascribed to their antioxidant ability, in cancer cells they may also function as pro-oxidants, acting as cytotoxic agents through the increase of ROS levels beyond critical thresholds limits [70]. Indeed, they act as antioxidants or radical sinkers depending on their concentration. They usually have pro-oxidant properties and act as cytotoxic agents at high doses (>50 µM), while exhibiting the opposite effects and acting as antioxidants at low doses. This dichotomy opens new challenges, given the difficulty of overturning the data obtained at bench to the bedside. Different ongoing clinical trials test the properties of the natural compounds derived from agri-food by-products in cancer treatment (Table 2).

Accordingly, some natural compounds, including aescin, morphine, paclitaxel, and vincristine have already been embraced by pharmaceutic companies, as indicated by their widespread use, and a plethora of beneficial natural compounds, useful in counteracting the cancer-related inflammation or cancer progression, now exist. Nevertheless, their mechanism of action remains pending. As depicted in the cartoon (Figure 2), different compounds derived from specific agri-food by-products are efficacious in different cancer types.

Table 3 summarizes the main compounds, along with the cancer types.

### 3.1. Breast Cancer

Breast cancer (BC) is a highly heterogeneous disease, classified into different categories according to sex steroid and epidermal growth factor (EGF) receptor-expression profiles. Estrogen receptors (ERs, α or β), progesterone receptors (PRs), and EGF receptors (EGFRs) are valuable markers for BC prognosis and survival [86]. About 20% of BC patients are affected by a particular subtype of BC, called triple negative (TNBC), characterized by the lack of ER, PR, and human epidermal growth factor receptor (HER2) expression, and generally accompanied by an aggressive clinical course and poor prognosis [87]. EGFR family members (EGFR/ErbB1/HER1, ErbB2/HER2, ErbB3/HER3, and ErbB4/HER4) are involved in BC progression, controlling different cell functions, such as differentiation, proliferation, survival, and migration. Moreover, a frequent EGFR overexpression can be detected in TNBC [88] and it is caused by a gene amplification in almost 25% of patients [89]. Currently, there are no targeted therapies approved for TNBC patients due to the absence of the typically targeted receptors. Therefore, the most employed treatments are based on a combination of surgery, chemotherapy, and radiation therapy [90,91]. Data, mainly from BC, lung cancer, and glioma have suggested many potential mechanisms related to aberrant EGFR signaling, such as high ligands and receptors expression, autocrine signaling loops, constitutive activation of EGFR mutants and crosstalk with other receptors [92]. Thus, the most employed therapies to target EGFRs are currently based on the use of neutralizing monoclonal antibodies (mAbs) or small-molecules tyrosine kinase inhibitors (TKIs) [93]. Nevertheless, patients often acquire resistance despite the high selectivity of mAbs, thus making their use limited. The resistance might be due to a constitutive activation of downstream effectors or an overexpression of other tyrosine kinase receptors. In addition, TKIs are not specific for EGFRs and might inhibit the activity of other tyrosine kinase receptors. TKI-acquired resistance mechanisms are commonly due to the T790M mutation, which prevents the TKI-binding to EGFRs, thus reducing the drug effectiveness [94]. The use of mAbs and TKIs is not routinely employed in BC, although there are a number of ongoing EGFR-targeted clinical trials (NCT05341492, NCT05177796, NCT02593175, NCT04395989; NCT03805399) in BC and TNBC.

A significant role in BC progression is played by cells that make up the TME and in particular by the tumor-associated macrophages (TAM) or leukocytes, considered a molecular signature of poor prognosis in BC patients. TAM can affect the invasiveness of epithelial BC cells, through the release of endothelins (ETs), promote the activation of inflammatory pathways through the activation of NF-κB [95], and trigger cell proliferation, endothelial cell activation, and dissolution of connective tissues [96,97]. Most BC patients exhibit increased levels of inflammasome components, such as interleukins (IL-1β and IL-18) compared with their normal counterparts. Again, elevated serum levels of IL-1βb in BC patients is associated with the establishment of inflammatory TME, tumor progression, and acquisition of a metastatic phenotype [98].

Recent data show the existence of a specific, lethal, and aggressive form of BC, called inflammatory BC (IBC) [99,100]. It exhibits overexpression of EGFRs, E-cadherin, and NF-κB [101] and give rise to distant metastasis [102]. In this BC subtype, the EGFR axis modulates the expression of cyclooxygenase-2 (COX-2), a molecule involved in the inflammatory response. As such, the IBC stemness might be forced and a loop between EGFR signaling, cancer cell stemness, and inflammation would sustain the cancer survival and progression [35].

As the link between chronic inflammation and BC progression is undeniable, a better understanding of the molecular basis of this connection is required for therapeutic purpose. In this regard, the agri-food by-products may be used as cochemotherapeutic agents to improve the pharmacological action of anticancer drugs in BC management and ameliorate the disease-associated inflammatory status. As already specified, these compounds might also exert pro-oxidant activity. Studies have been carried out by combining natural compounds with approved drugs. Hesperidin, deriving from the ethanolic extract of lime peels (*Citrus aurantifolia*), increases the cytotoxic effect of the drug, blocks the cell cycle, and induces apoptosis in BC-derived cells when used in combination with doxorubicin [103]. Nobiletin and naringenin, two flavonoids isolated from citrus peels, increase the cytotoxic activity of doxorubicin [104,105] and promote BC cell death [72]. However, agri-food by-products seem to be effective also when used alone. Tangeretin, derived from mandarin peel (*Citrus reticulata*) and lime (*Citrus aurantifolia*) extract and the ethanol extract of avocado seeds (*Persea Americana*) has a cytotoxic effect and inhibits BC cell proliferation, inducing the G1 blockade of the cell cycle [106]. PhCs, derived from *annurca* apple, mitigates oxidative stress, induces antiproliferative effects, and promotes apoptosis through the inhibition of MAPK activity, the upregulation of p53 and p21, as well as the downregulation of cyclin D1 [74]. Quercetin inhibits proliferation, survival, and differentiation of BC cells. By impairing the VEGFR-2/p-EGFR and p-PI3K/Akt/p-GSK-3 pathways, quercetin inhibits epithelial-mesenchymal transition (ETM), proliferation, migration, and invasion when encapsulated in nanoparticles [76]. It downregulates leptin gene expression [107], survivin [108], and modulates the Bax-Bcl2 pathway, thus promoting necroaptoptosis [109] and inhibiting cell growth [110]. Quercetin also exerts a negative effect on angiogenesis through the repression of VEGF, its receptor VEGFR2, and the calcineurin pathway [111,112]. Finally, it impairs the p38 mitogen-activated protein kinase (p38MAPK) and PI3K/Akt pathways [113]. Again, the effect of the green tea catechin, also found in apple peel, epigallocatechin-3-gallate (EGCG), has been well investigated in BC. Catechins belong to the family of polyphenols and are present in green and black tea, red wine, and chocolate. They are composed of a basic 2-phenylchromone structure and characterized by the di- or tri-hydroxyl group substitution of the B ring, the 2,3-position isomer of the C ring, and the presence of a galloyl group at the 3-position of the C ring [114]. Depending on the different conformational changes, they include EGCG, epicatechin, gallocatechin, epigallocatechin, catechin gallate, epicatechin gallate (ECG), gallocatechin gallate, and catechin [115]. Among them, EGCG and ECG account for up to 76% of catechins in the tea plant [116]. EGCG exhibits antiproliferative, anti-inflammatory, antimetastatic, and apoptotic effects acting through the downregulation of ERα, PI3K/Akt signaling, cyclin D1, and β-catenin and the upregulation of Bax, p53, caspase-3 and -9, and PTEN [117,118,119,120,121]. EGCG shows anti-inflammatory and antiangiogenic effects through the activation of the Nrf2 pathway [122] and the inhibition the NF-kB and VEGF pathways [123]. It also shows epigenetic effects, since it may impair DNA methylation and induce histone modifications, reducing DNMT1, HDAC1, and methyl CpG-binding protein 2 (MeCP2) expression [124]. Lastly, EGCG has an important role in BC-TME, since it inhibits TAM infiltration [125]. Resveratrol, found in grapes, berries, and peanuts [126,127,128] impairs the molecular pathways involved in cell proliferation, death, invasiveness, epigenetic modifications, and chemosensitization [129]. Resveratrol increases the expression of BRCA1, p53, and p21 and reduces the expression of ERα, cyclin D1, and cyclin B1 [130]. It inhibits the signaling involving Wnt and telomerase and hence cell proliferation [131,132] and downregulates the insulin-like growth factor (IGF), epidermal growth factor (EGF), MAPK, and PI3K/Akt/mTOR signaling thereby impairing cell invasion and proliferation [133]. Resveratrol reduces the phosphorylated status of Akt and upregulates phosphatase and tensin homolog (PTEN) expression, leading to the suppression of the PI3K/Akt/mTOR pathway, which is usually overactive in cancer cells [134]. Figure 3 summarizes the main biological effects, or the events modulated by the different natural compounds in BC.

Currently, the research of natural compounds exhibiting antioxidant activity with few side effects has made the leap from research laboratories to the pharmaceutical industry, giving it greater importance in BC, because of the great involvement of the proinflammatory components, such as cytokines, (TNF-α and IL-6) that promote BC cell proliferation and metastasis [135].

### 3.2. Prostate Cancer

PC arises from the gland cells and almost all PCs are adenocarcinomas. This definition also includes small cell carcinomas, neuroendocrine PC (NEPC), transitional cell carcinomas, and sarcomas, which initiate in prostate to metastasize in other organs and tissues. Currently, different biomarkers can be used to guide treatment decisions. In addition to PTEN loss and high Ki-67 labelling index, which are strongly associated with adverse clinical outcomes [136], more specific biomarkers might be used for advanced forms of PC. Among them, AR, and its splicing variant AR-V7, synaptophysin, chromogranin, or CD56 (for NEPC), homologous recombination deficiency (HRD), and prostate-specific membrane antigens (PSMAs) are employed for novel therapeutic approaches, including poly ADP ribose polymerase (PARP) inhibitors. In a small percentage of PC patients there is also an alteration of EGFRs [137], which represents a positive inclusion criterion in different clinical trials, including NCT00148772, NCT04776889, NCT00953576, NCT00103194, NCT00239291, and NCT00023634. However, while localized PC can be cured by local treatment and surgically removed, advanced PC is often managed with the androgen deprivation therapy (ADT), which might fail resulting in PC progression to castration resistant (CRPC), and metastatic disease [138].

By impairing the activity of type I and type II 5-α-reductases, some agri-food by-products affect the testosterone synthesis and prevent the synthesis of the more potent ligand, dihydrotestosterone (DHT) [139]. It has been, hence proposed the combo use of these compounds with standard ADT to improve the efficacy of treatment and delay the onset of CRPC. A clinical trial has shown that the extract from licorice root (*Glycyrrhizia glabra*) significantly decreases testosterone levels in healthy female volunteers, by targeting two enzymes, the 17,20-lyase and 17 βhydroxysteroid dehydrogenase, involved in androgen synthesis [140].

Drugs inhibiting the expression of AR would represent another therapeutic strategy to fight PC. In this regard, it has been reported that EGCG downregulates the AR expression and inhibits the receptor nuclear translocation in PC cells through the inhibition of NF-kB activity [141,142,143]. EGCG has been suggested to decrease NF-kB activity by suppressing its acetylation [144]. Similarly, other polyphenols derived from green tea, grape seeds, and pomegranates target NF-kb and control the growth of PC cells in vitro and in vivo [145]. Catechins, found in green tea can also impair the progression of PC and may be of particular benefit to men affected by PC with a low Gleason’s score who are usually placed on active surveillance. This might be due to the gut microbiome that acts as an important mediator in regulating the bioavailability of catechins and absorption of bioactive phenolic metabolites. Bacteria, indeed, convert ECGC in different metabolites, thus contributing to ameliorate the PC patients conditions, regulating the inflammation, the hormone levels, and other known and unknown pathways [146].

Again, isoflavones downregulate AR expression, through the inhibition of Akt and FOXO3a phosphorylation in AR-expressing PC-derived LAPC-4 cells, and upregulate AR expression in LNCaP cells harboring a threonine to alanine (T877A) mutation of the AR [147]. Despite the reported dichotomic role, it is undeniable that isoflavones exert an effect on AR expression and warrant further investigations.

In general, agri-food by-products exert antiproliferative and antimetastatic effects in PC. In this context, we can mention the hesperidin and its antitumoral potential in inhibiting migration, invasion, and colony formation and inducing the G2/M phase cell cycle block, apoptosis, and necrosis. Again, nobiletin is considered a chemopreventive agent in the treatment of PC, able to inhibit transformation, proliferation, and cellular invasion, impairing NF-kB signaling, VEGF expression, and AKT phosphorylation [82]. Tangeretin causes apoptosis of both dependent and independent PC cells through the regulation of caspase-9 and -3, Bax, and Bcl-2. It is also involved in EMT through the regulation of proteins such as Vimentin, CD44, E-cadherin, and cytokeratin-19 [83]. Quercetin inhibits proliferation of PC cells by decreasing the phosphorylation of EGFRs and the expression of EGF and EGFR mRNA. In addition, it inhibits the p-Akt/GSK3 signaling by acting on EGFRs. Quercetin also regulates proapoptotic and antiapoptotic proteins, reducing Bcl-2 levels and increasing Bax expression [84]. All the mentioned natural compounds can impair oxidative stress and the inflammatory status of PC cells or exhibit cytotoxic activity depending on the concentration of use. The main biological effects or cellular events are presented in Figure 4.

New strategies to target PC progression are envisaged and several natural products selectively target different molecules and signaling pathways implicated in PC development, progression, and inflammatory response [148]. Many of them have been tested in in vitro and in vivo, while some clinical trials have been so far conducted or are currently ongoing [149]. Further studies are needed for a better understanding of the agri-food by-products as adjuvants in counteracting the PC inflammatory network.

### 3.3. Colorectal Cancer

Colorectal cancer (CRC) is the third most prevalent cancer [150] with a higher death rate for males compared to females. Standard cancer therapy typically involves the triple regimen of surgery, chemotherapy, and radiation treatment. Unfortunately, current therapies are not exempt from side effects and the acquisition of drug resistance. Several epidemiological studies have demonstrated the association of colon cancer with dietary habits such as low fiber intake, high-fat diet, and low calcium and micronutrient intake [151,152,153]. Thus, there is an increasing trend to focus on natural sources, including plants and fruits, for searching novel anticancer agents. Results obtained in preclinical and clinical models suggest that bioactive compounds derived from fruits and vegetables, including their waste products, might help in the prevention and treatment of CRC [154,155,156].

Some studies have analyzed the antitumoral potential of apple peel extracts in colon cancer cells showing that quercetin-rich peel juice extract diminished DNA damage and ROS level in CRC cells more than apple pomace extraction juice [157]. Thus, we can deduce that the bioactive molecules responsible for the anticancer activity are in the apple peel [158].

We still have few data regarding the effect of apple waste products on colon cancer, but the findings may suggest that triterpenoids may be partially responsible for apple peel’s antitumoral activity in colon cancer along with other bioactive compounds such as flavonoids [159]. Quercetin also induces apoptosis in colon cancer cells through the regulation of p53, suppresses carcinogenesis in rat models of colon cancer through the inhibition of enzymes involved in inflammation (cyclooxygenases COX-1 and COX-2, as well as iNOS) [160], and when used in combinatorial approaches with doxorubicin is able to arrest the cell cycle in the G2/M phase [161].

Similarly, another molecule potentially effective in the reduction of the inflammation is the EGCG which is able to decrease the COX-2 activity in in vivo models. It could be a promising molecule for colon cancer treatments, given its capability of reducing cell growth, arresting the cell cycle in G1 phase, and inducing the apoptosis by the activation of caspase-3 and -9, with a critical impact on EGFR/ERK and EGFR/Akt signaling pathways [162,163].

Peel and seeds of citrus species contain phytochemicals promising to be employed in CRC therapy protocols as chemopreventive and chemotherapeutic agents. Citrus has been shown to modulate the inflammation and immune responses and suppress cellular oxidative stress, promoting Nrf2 transcriptional activity and reducing NF-kB, p53 and STAT3 transcriptional activity [164]. Citrus peels are rich in polymethoxyflavones (PMFs), whose role in gastrointestinal cancer have been widely investigated in several in vitro and in vivo models. The most effective PMFs, sinesentin, nobiletin, and tangeretin, are able to inhibit the proliferation of human CRC cells in 3D models, arrest the cell cycle in G1 or G2 phases through the upregulation of p21 and p27, as well as the inhibition of CDK-2 and CDK-4 [165]. They also promote the apoptosis of colon cancer cells [166,167], without affecting the surrounding normal cells [106].

The PMFs induce human cancer cell apoptosis through different mechanisms and signaling pathways also depending on the cell type. In gastric adenocarcinoma cells, for example, they promote an autophagic cell death inducing cytoplasmic vacuoles and autophagosomes. This process involves the PI3K/Akt/mTOR pathway downregulation, the p21 upregulation, the activation of Beclin 1 and LC3B, and the activation of MAPKs [168]. In addition, its metabolites, such as 30-demethylnobiletin (30-DMN), 40-demethylnobiletin (40-DMN), 30,40-didemethylnobiletin (30,40-DMN) are promising antitumoral agents, since they significantly inhibit cell proliferation, cause cell-cycle arrest, induce apoptosis, and negatively modulate the signaling cascade linked to cell proliferation and cell death [169]. Current studies are involved to deepen the beneficial synergistic effects exerted by the concomitant administration of nobiletin and its metabolites or nobiletin and other drugs which would seem very hopeful in inhibiting inflammation, cell cycle progression, angiogenesis, and metastasis [170].

Recently, a class of hydroxylated PMFs, derived from the long-term storage of citrus peel, has been isolated. These compounds have a strong inhibitory effect on cell proliferation associated with the ability in modulating key signaling proteins involved in cell proliferation and apoptosis, such as p21Cip1/Waf1, CDK-2, CDK-4, phospho-Rb, Mcl-1, caspases-3 and -8, and PARP [171].

Experiments performed in in vivo models show that a long-term dietary consumption of hydroxylated PMFs resulted in the reduction of colon cancer cell proliferation and inhibition of the inflammation through the downregulation of Wnt/β-catenin and EGFR signaling pathways as well as the activation of STAT3 and NF-kB transcription factors, thus blocking the expression of iNOS, COX-2 VEGF and MMP-9 [172]. In addition, citrus seeds can be a promising source of bioactive compounds that may help to prevent colon disease progression and malignancy. Limonin and limonin glucoside isolated from seeds of *Citrus reticulata* inhibit the colon cancer cell proliferation inducing the cell cycle block in G0/G1 phase, suppressing the CDK4/6 and cyclin D3 expression and inducing the apoptosis, through the DNA fragmentation and the activation of caspase-3 [173].

The chemopreventive and chemotherapeutic potential of the dietary administration of citrus limonoids have been studied in different in vivo models. In particular, dietary feeding of limonin and another limonoid compounds, obacunone, significantly inhibited the formation of earliest neoplastic lesions and reduced the frequency of colon adenocarcinoma in mice [174]. Limonin is being increasingly considered for its immune-modulating effects. It mitigates CRC development by enhancing the adaptive immune responses (CD8+ Cytotoxic T lymphocytes (CTL) cells, CD4+ and CD19 cells) and restoring the innate immune responses to normal conditions [175].

It increases the natural killer (NK) cells number, contributing to the acquisition of the adaptive immunity in CRC mice models [175,176]. Moreover, limonin administration downregulates the inflammation reducing IL-1β induced proinflammatory cytokine levels such as iNOS, COX-2, PGE2, NO, TNF-α, and IL-6, weakens oxidative stress, and enhances the endogenous antioxidation defenses in mice by regulating Nrf2 and SOD2 [175,177].

The liaison between CRC cells and immune cells is highly complex and chronic inflammatory status is one of the principal features of colorectal and colitis-associated tumors, with 15%–20% of cancers related to an underlying chronic inflammation process, modulated by immune innate and adaptive cell infiltration and immune cells present in TME [38,178]. Significant progresses have been made in understanding the role of the immune system in driving the development of CRC. The studies so far conducted have revealed two trends, the identification of immune cell markers to predict cancer outcomes and modulate the inflammatory status of CRC. The combinatorial therapies employing cytotoxic drugs and multiple immunotherapeutic modulators represent, indeed, the most promising approach. As such, citrus limonoids appear very useful to prevent and treat CRC.

All the mentioned natural compounds impair oxidative stress and the inflammatory status or exhibit cytotoxic activity in CRC cells (Figure 5). Here again, hard investigations would establish the concentration’s threshold that exerts the different effects.

## 4. EGFRs as a Putative Target of Agri-Food By-Products

The pleiotropic effects of agri-food by-products and their cellular promiscuous activity are quite relevant and suggest that multitargeting therapy can be a future strategy for cancer treatment. However, this type of research is limited by the fact that most previous studies were more likely to analyze the effects exerted by these natural compounds in cancer cells rather than their intracellular targets. The modern computational methodologies have recently provided useful information in the discovery of novel and suitable small molecules for protein targeting [179]. In this context, the quantitative structure–activity relationship (QSAR) and molecular docking are the two most used methods for this application [180,181] given their ability to predict the property of a chemical substance, the function of its structure and to show the bonds between the protein and the small molecules in a three-dimensional plot.

Recent data have shown that small molecules from natural sources, particularly PhCs and flavonoids, may have a better EGFR inhibitory activity compared to commercial drugs [182]. PhCs exhibit high-binding affinity for the EGFR tyrosine kinase, such as that of other selective inhibitors of the receptor kinase activity in molecular docking models [183]. Accordingly, recent data have shown that curcumin, a polyphenol extracted from *Curcuma longa*, affects the EGFR tyrosine kinase autophosphorylation [184] and induces the EGFR downregulation [185] in a dose- and time-dependent manner in different types of cancer cells. Again, ECGC inhibits the EGFR activity by disturbing the “lipid raft” and therefore impairing EGFR dimerization and activation [186]. Furthermore, ECGC, weakens the Tyr1068 EGFR phosphorylation so affecting the receptor recycling after lipid raft alteration [187], inhibiting the downstream effectors ERK and Akt and the transcriptional activity of AP-1, c-fos, NF-kB, and cyclin D1 promoters in colon cancer cells [162].

Luteolin inhibits the EGFR-dependent protein kinases and EGFR autophosphorylation in metastatic PC cells [188]. It synergistically enhances the effect of lapatinib, which inhibits HER2 and EGFR, and impairs Akt- and ERK-signaling pathways, resulting in a decrease of BC cell proliferation [189]. Agri-food by-products derivatives also impair motility, invasiveness, or EMT in different types of cancer cells, through EGFR signaling modulation. Particularly, quercetin inhibits the EGF-induced activation of EGFRs, PI3K and Akt, thus inhibiting migration, invasiveness and EMT of PC cells [76,190,191]. Again, quercetin 3-O-glucoside reduces the pancreatic cancer cells migration through inhibition of the EGFR-dependent signaling cascade (FAK, Akt, MEK 1/2, ERK) [192]. Thus, it is arguable that the natural products directly act on the EGFR intracellular domain, on one hand. On the other, they impair EGFR diffusion and dimerization because of their ability to interfere in cell membrane plasticity (Figure 6A). Furthermore, these natural derivatives also downregulate EGFR expression through its ubiquitination and degradation (Figure 6B), albeit the fine regulation of the degradation process is still to be determined. Through these mechanisms, polyphenols derived from agri-food by-products impair EMT and inhibit cell proliferation and migration, preferentially by deactivating the AKT and ERK-dependent signaling cascades (Figure 6C).

As stated before, EGFRs are overexpressed in PC, and their expression levels correlate with tumor progression [193]. Increased expression and signaling of EGFRs and ErbB2 in PC specimens are associated with a more aggressive clinical behavior and a poor prognosis [194,195,196]. Indeed, metastatic lesions from hormone-refractory PC always express EGFRs, suggesting a key role for this receptor in the aggressiveness of this type of cancer [197]. Again, EGFR/ErbB2 heterodimers increase the metastatic potential of BC cell lines [198] and the BC basal type, often classified as TNBC, is characterized by EGFR overexpression that results in chemo- and radio-resistance to chemotherapy and radiation treatment, hence leading to poor prognosis and survival [199,200]. The role of EGFR in CRC establishment and progression is undebatable, as the receptor expression level progressively increases with malignant transformation from normal colon tissues, through adenoma, to the poorly differentiated and metastatic cancer [201], with lympho-vascular infiltration [202]. EGFRs are overexpressed in CRC cell lines and can be detected in about 70% of CRC tumors [203].

In summary, given the importance of EGFRs as biomarkers in different types of cancer, including BC, PC, and CRC, the findings discussed here might be relevant to conceive new therapeutic strategies. EGFR-targeted therapy still represents a big challenge in clinical oncology, mainly because of its intrinsic or acquired resistance, which often limits the efficacy of targeted therapies [204,205]. For this reason, new easily available natural derivatives might be useful to prevent cancer-related inflammation and foster the efficacy of currently used chemotherapeutic agents.

## 5. Concluding Remarks

Research on the recovery of agri-food by-products has had a significant boom in recent years. It is well established that the food industry generates many by-products worldwide with a high content of compounds that are discarded from the food transformation process, but that can be useful as a raw material in the production of drugs. In addition, the extraction of these compounds is performed using sustainable techniques, such as hydroalcoholic extraction in the case of the PhCs (hesperidin, nobiletin, and tangeretin) or hydro-distillation in the case of citrus peel. The high content of organic acids, sugars, and PhCs contain antibacterial, antifungal, and anti-inflammatory properties, thus the agri-food waste can also improve the bioavailability of different drugs. Finally, the cost of agri-food might be considered an attractive option for low-cost drug production. Its application by pharmaceutical companies might represent a more sustainable solution, especially in low-income countries where the access to healthcare and cancer treatment is often limited.

These properties open a new arena in the food sector based on the development of novel functional foods. Some of these compounds, such as citrus peel extracts, can be used in their natural form and mixed in nutraceutical formulations. Others, such as tangeretin and quercetin, are modified and encapsulated in nanotechnological formulations to improve their stability, because of their lipophilic nature and low solubility.

The discovery of natural substances able to interfere at various degrees in growth factor-elicited cancer aggressiveness might be beneficial for new approaches of neoplastic diseases, especially considering that these compounds may be used in cancer prevention for limiting the inflammatory process leading to cancer and other chronic diseases, and in cancer treatment due to their cytotoxic activity. Data reported in this review have shown that many agri-food by-products selectively target cancer cells through many mechanisms in in vitro and in vivo models. However, further investigation is required to determine the effective functionality and safety of these compounds, and their potential in mono- or combinatorial therapy, especially in regard to the concentrations needed for their effective action. Nevertheless, the limited number of clinical studies so far documented indicates that additional studies are required to evaluate the anticancer activity of these compounds in experimental animal studies and clinical trials. Moreover, their impact in the receptor tyrosine kinase signaling (e.g., EGFR, IGFR, VEGF, or nerve growth factor (NGF) receptors) warrants further investigations, because of the role of these receptors and their intersection with the so-called nuclear receptors in many human proliferative diseases, including breast, prostate, and colon cancers [206,207,208,209,210,211]. Further studies would confirm the current outcomes and may ultimately lead to new treatment options.

## Figures and Tables

**Figure 1 cancers-14-05517-f001:**
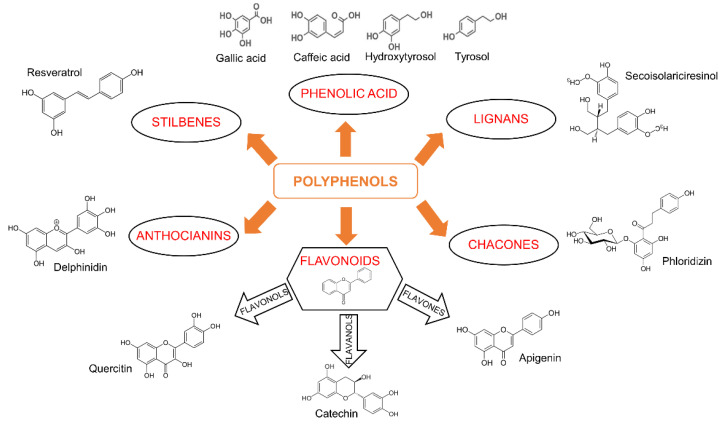
Classification of polyphenols and chemical structures of some representative phenolic compounds (PhCs). Phenolic acids are nonflavonoid PhCs that give origin to different derivatives. The most abundant class of PhCs is represented by flavonoids, categorized into different subclasses (flavones (apigenin), isoflavones, flavanols (catechins), flavonols (quercitin), flavanones, anthocyanins, and proanthocyanidins). Structural differences distinguish the anthocianins and the stilbenes, which include compounds such as resveratrol. A small amount of PhCs is represented by lignans and chacones.

**Figure 2 cancers-14-05517-f002:**
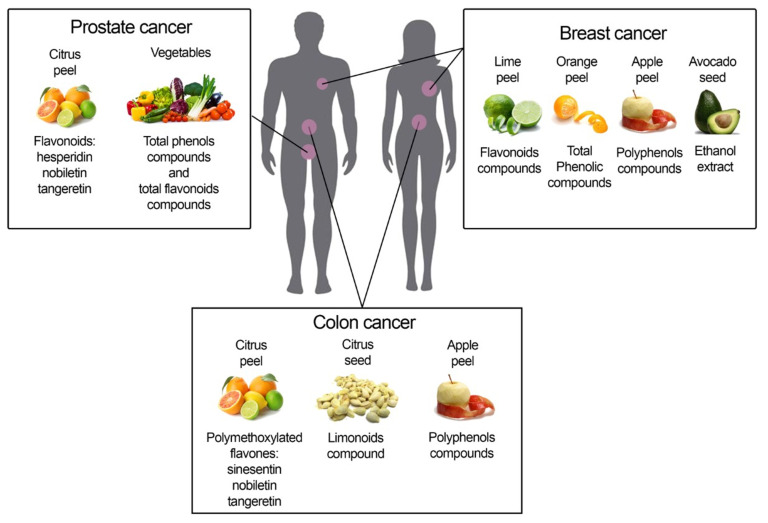
Depicts agri-food by-products and their derived molecules able to interfere with prostate, breast, and colon cancer.

**Figure 3 cancers-14-05517-f003:**
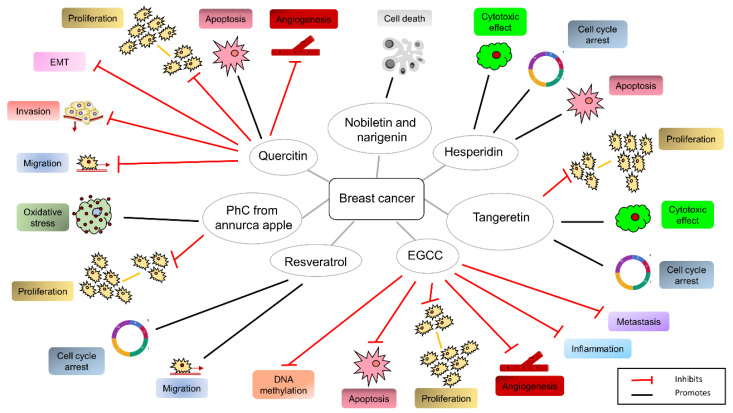
The biological effects and the events modulated by the agri-food by-products derivatives in breast cancer.

**Figure 4 cancers-14-05517-f004:**
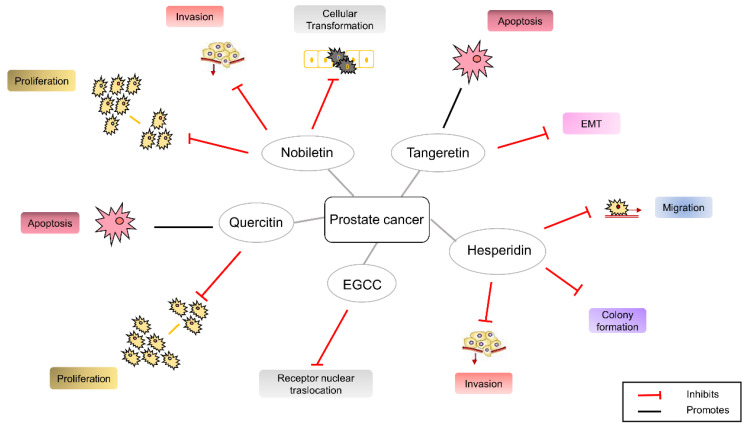
The biological effects modulated by the agri-food by-products derived-natural compounds in prostate cancer.

**Figure 5 cancers-14-05517-f005:**
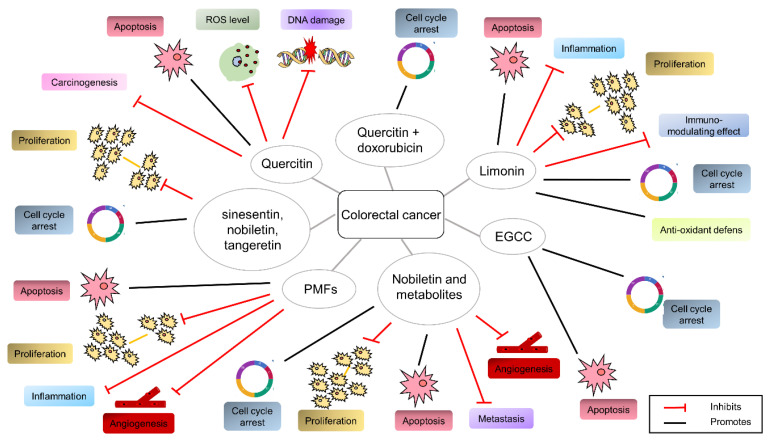
The agri-food by-products derived from natural compounds exert the illustrated effects in colorectal cancer.

**Figure 6 cancers-14-05517-f006:**
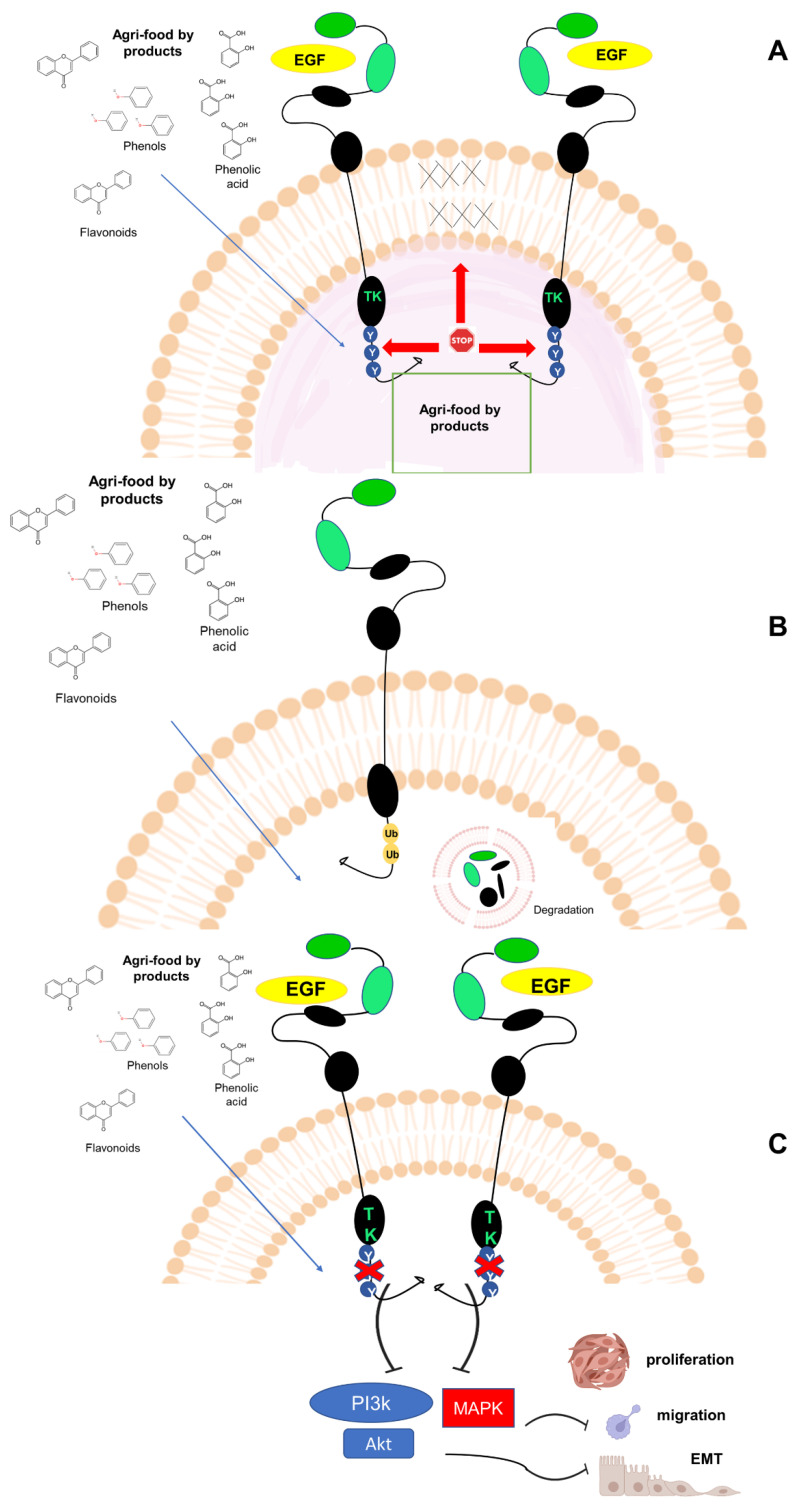
The putative mechanisms of action of agri-food by-products derivatives on EGFRs. Natural derivatives (e.g., flavonoids, phenolic acids, and phenols in (**A**–**C**) affect the EGFR signaling through different mechanisms. (**A**) They directly impair EGFR tyrosine autophosphorylation and its tyrosine kinase activity (TK) or indirectly affect EGFR dimerization by interfering in the cellular membrane bilayer fluidity. The membrane stiffness prevents EGFR diffusion and dimerization, with a negative effect on the TK resulting activation. (**B**) Polyphenols downregulate EGFR expression through protein ubiquitination and degradation. (**C**) By inhibiting EGFR-TK domain, natural compounds impair the EGF-induced PI3K/Akt and -MAPK signaling cascades. These effects halt cellular migration, proliferation, and EMT.

**Table 1 cancers-14-05517-t001:** Compounds isolated from fruits or vegetables along with the bioactivity and the type of the diseases where they exert antioxidant activity.

Agri-Food By-Product	Origin	Compound	Bioactivity	Disease	Reference
Orange peel	*Citrus sinensis* L.	Hesperidin	Gastroprotection	Gastrointestinal and duodenal disease (Ethanol-induced peptic ulcers)	[11]
Avocado seed	*Persea Americana Mill. (Lauraaceae)*	Ethanol extract of *Persea americana* seeds	Antiendometriosis effect	Tamoxifen induced endometrial hyperplasia and endometriosis	[53]
Olive oil wastewater of mill	*Olive*	Hydroxytyrosol	Antioxidant and anti-inflammatory effect	Corneal cells inflammation and dry eyes disorders	[24]
Pineapple peel, core, and leaves	*Ananascomosus*	Bromelain	Antiedematous, antithrombotic, anti-inflammatory	Rhinitis, rhinosinusitis and chronic rhinositusitis; intestinal inflammation and Chron’s disease; asthma; osteoarthritis and rheumatoid arthritis; cardiovascular diseases; thrombophlebitis; angina pectoris	[54,55,56,57,58,59,60]

**Table 2 cancers-14-05517-t002:** Emerging interventions based on the use of agri-food by-products derivatives in clinical trials for the treatment of patients with different types of cancer.

Agri-Food By-Products Derivative	Type of Cancer	Intervention	Status/Phase	Identifier Number
Bromelain	Breast, colon, ovary, uterus, cervix, lung	Bromelain and Comosain	Phase II	NCT02340845
Bromelain	Breast	Bromelain and Bosswellia Serrata	Phase II	NCT04669119
Quercitin	Prostate	Quercitin and Genistein	Recruiting	NCT01538316
Quercitin	Childhood cancer	Quercitin and Fisetin	Phase II	NCT04733534
Quercitin	Prostate	Quercitin and green tea extract	Phase I	NCT01912820
Quercitin	Colorectal	Quercitin, curcumin and rutin	Study completed	NCT00003365
Luteolin	Tongue, mouth, head and neck	Luteolin	Early phase I	NCT03288298
Polyphenols (hesperidin, curcumin, resveratrol and ellagic acid)	Breast	Polyphenols	Study completed	NCT03482401
Bromelain	Breast cancer	Combinatorial therapy and Bromelain	Phase IV	NCT001609001
Curcumin	Colonic cancer, metastasis	Curcumin and chemotherapy	Phase I, Phase II	NCT01490996
Curcumin	Advanced and metastatic breast cancer	Curcumin and Paclitexel	Phase II	NCT03072992
Curcumin	Breast cancer	Curcumin	Phase II	NCT01740323
Curcumin	Multiple myeloma	Curcumin	Phase II	NCT01269203

**Table 3 cancers-14-05517-t003:** Examples of the most investigated compounds derived from fruits or vegetables along with the type of cancer where they exert an effect.

Compound	Type of Cancer	Reference
Phenolic extracts from orange peels	Colon cancer	[71]
Bromelain	Breast cancer, Melanoma	[55]
Naringenin	Breast cancer	[72]
Bromelain	Gastric cancer	[73]
Apple peel and flesh extracts	Breast cancer	[74]
Ethanol extract from avocado seeds	Breast cancer	[75]
Quercitin	Breast cancer	[76]
Bromelain	Melanoma	[77]
Luteolin	Melanoma	[78]
Naringenin	Melanoma	[79]
Elleagic acid	Melanoma	[80]
Hesperidin	Prostate cancer	[81]
Nobelitin	Prostate cancer	[82]
Tangeretin	Prostate cancer	[83]
Quercitin	Prostate cancer	[84,85]

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
