# Peer review of "Agri-Food By-Products in Cancer: New Targets and Strategies"

_cancers, 2022, doi:10.3390/cancers14225517_

Round 1
Reviewer 1 Report
Thank you for providing me with the opportunity to review the manuscript entitled " Agri-Food By-products in cancer: new targets and strategies".
Manuscript Cancers-1956151 is a review article and its aim is the to summarize some healthy effects of agri-food by-products derivatives. The research on the recovery of agrifood by-products had a significant boom in recent years and it is a developing topic.
This is a study with clear and interesting results that would add to the evidence already published. The text is clear and easy to read. The results are clearly presented. The conclusions are consistent with presented evidence and arguments. References are up to date and complete.
Carefully reread the fruit names.
For example:
Line 56: “Citrus sinensis” is “Citrus sinensis" (italic font)
Line 57: “Citrus limon” is “Citrus limon” (italic font)
Line 57: “Citrus aurantiifolia” is “Citrus aurantiifolia” (italic font)
Line 58: “annurca” is “Annurca”
Author Response
Rebuttal letter
Dear Editor and reviewers,
Many thanks for forwarding such positive comments on our manuscript and the appropriate and interesting suggestions.
Below we have detailed how we have addressed the major comments raised by the Reviewers and discussed some other minor textual changes that we have incorporated for increased clarity.
We trust that this revised version of our MS will be acceptable for publication in “Cancers”.
Sincerely,
Marzia Di Donato, PhD
Assistant Professor of General Pathology
Reviewer 1
Thank you for providing me with the opportunity to review the manuscript entitled " Agri-Food By-products in cancer: new targets and strategies".
Manuscript Cancers-1956151 is a review article and its aim is the to summarize some healthy effects of agri-food by-products derivatives. The research on the recovery of agrifood by-products had a significant boom in recent years and it is a developing topic.
This is a study with clear and interesting results that would add to the evidence already published. The text is clear and easy to read. The results are clearly presented. The conclusions are consistent with presented evidence and arguments. References are up to date and complete.
Carefully reread the fruit names.
For example:
Line 56: “Citrus sinensis” is “Citrus sinensis" (italic font)
Line 57: “Citrus limon” is “Citrus limon” (italic font)
Line 57: “Citrus aurantiifolia” is “Citrus aurantiifolia” (italic font)
Line 58: “annurca” is “Annurca”
We thank the Referee, and we appreciate his/her suggestion. We have rewritten the fruit names by using the italic font.
Reviewer 2 Report
The manuscript is well written. The author offers an original approach in summarizing the effects of Agri-Food by-product derivatives in human diseases, including different types of cancer. I suggest a small modification to table 1 also specifying: "Byproduct, Origin, Compounds, Bioactivity", in order to give the reader greater usability of these Agri-Foods. Overall, the manuscript is produced at a high scientific level, it contains a series of conclusions of practical interest.
Author Response
Reviewer 2
The manuscript is well written. The author offers an original approach in summarizing the effects of Agri-Food by-product derivatives in human diseases, including different types of cancer. I suggest a small modification to table 1 also specifying: "Byproduct, Origin, Compounds, Bioactivity", in order to give the reader greater usability of these Agri-Foods. Overall, the manuscript is produced at a high scientific level, it contains a series of conclusions of practical interest.
We thank the Referee for his/her useful suggestions and nice comments. We have modified table 1 as suggested.
Reviewer 3 Report
Although the manuscript is well written and organized, it should be further updated to be published in Cancers as follows:
Major Points
1. The compounds from agri-food by products play a role as antioxidant/radical sinker that is double edge sword in prevention of tumorigenesis and cancer therapy. In order to publish their manuscript in a higher-level investigative journal than a nutrition-related general journal, the authors are strongly recommended to provide a specific (or general) mechanism of action for each chemical (both cancer prevention and cancer treatment), such as nobiletin, naringenin, tangeretin, catechin epicatechin and epigallocatechin etc.. Schematic diagrams of each compound in relation with the EGFR signaling pathway may be very helpful for audience in Cancer Research field. In addition, authors need to provide a proposed mechanism of Natural flavonoids, (poly)phenol compounds in protection of tumorigenesis and cancer cell specific killing (if possible) with grouping of them.
2. Although the authors described that the mechanism of action of PC and flavonoids from agri-food by products is unclear or occurs through many mechanisms, these compounds act as a radical scavenger/sink in the cell membrane, and also act as a radical buster in the active radical generation cascade, and the innate different effect between normal cells and cancer cells (activation of EGFR? or other signaling molecules) appears as a difference in sensitivity to radical/ROS. Again, the author needs to reinforce this detailed mechanism of action of these products.
3. EGFR is increased in various breast cancer subtypes, which is strongly associated with poor prognosis. The author limited the description only to IBC (minor population of BC with heterogeneity), author need to reinforce the mechanism of action for agri-food byproducts in a broader range of breast cancers with increased EGFR level.
In addition, Line 173-174 Regarding IBC, author needs to look for more recent research articles.
4. Author needs to address the clinical trial status for various diseases of various agri-food byproduct originated natural compounds.
Minor
1. Quercetin has been studied widely in CRC, and the following references will be used to further enrich the content.
Rather RA, Bhagat M. Quercetin as an innovative therapeutic tool for cancer chemoprevention: Molecular mechanisms and implications in human health. Cancer Med 2020; 9: 9181-9192 [PMID: 31568659 DOI: 10.1002/cam4.1411]
Kashyap D, Mittal S, Sak K, Singhal P, Tuli HS. Molecular mechanisms of action of quercetin in cancer: recent advances. Tumour Biol 2016; 37: 12927-12939 [PMID: 27448306 DOI: 10.1007/s13277-016-5184-x]
Sharifi-Rad M, Pezzani R, Redaelli M, Zorzan M, Imran M, Ahmed Khalil A, Salehi B, Sharopov F, Cho WC, Sharifi-Rad J. Preclinical Pharmacological Activities of Epigallocatechin-3-gallate in Signaling Pathways: An Update on Cancer. Molecules 2020; 25 [PMID: 31979082 DOI: 10.3390/molecules25030467]
2. Regarding Nobiletin and Catechin the authors need more discussion and include the following references not merely adding them.
“Nobiletin and Derivatives: Functional Compounds from Citrus Fruit Peel for Colon Cancer Chemoprevention” Cancers 2020
“Effects of Green Tea Catechins on Prostate Cancer Chemoprevention: The Role of the Gut Microbiome” Cancers 2022
3. Line 55 The ref 14 research was carried out in streptozotocin induced diabetic mice. This animal model is T1DM not T2DM. Author need to check it and look for the agri-food byproduct originated natural compounds on T2DM research.
4. Check the format of Tables and figures. Typos.
5. Table 1 Hydroxytyrosol, biophenol in the wastewater of mill seems to be Hydroxytyrosol, biophenol in olive oil wastewater of mill (OMWW)
6. Line 301 Ref 151 is unsuitable.
7. Line 305 Needs reference
8. There is lots of missed references in conclusion section (174, 163, 164, 165 167 175,168 etc),
9. Author needs to discuss the specific/common lethal mechanism of Nobiletin, naringenin, Tangeretin etc
10. Add the difference between catechin epicatechin and epigallocatechin
Author Response
Reviewer 3
Although the manuscript is well written and organized, it should be further updated to be published in Cancers as follows:
We thank the Referee for his/her concerns and suggestions.
Major Points
- The compounds from agri-food by products play a role as antioxidant/radical sinker that is double edge sword in prevention of tumorigenesis and cancer therapy.
It is a very intriguing point. We took a cue from this reviewer’s clarification, and we have now introduced this concept already in the introduction section and then, more extensively, in paragraph 3. In this regard, we have also changed the title of paragraph 3 that now is “Agri-food by-products in cancer: antioxidant or cytotoxic agents?”. Thus, in our opinion, it seems more appealing.
In order to publish their manuscript in a higher-level investigative journal than a nutrition-related general journal, the authors are strongly recommended to provide a specific (or general) mechanism of action for each chemical (both cancer prevention and cancer treatment), such as nobiletin, naringenin, tangeretin, catechin epicatechin and epigallocatechin etc.. Schematic diagrams of each compound in relation with the EGFR signaling pathway may be very helpful for audience in Cancer Research field.
Thank you for this suggestion. We have now described the putative mechanisms by which the different natural compounds can impair EGFR signaling. We have introduced in the new version of our review a new paragraph (Paragraph 4), titled “EGFR as putative target”. In this new section, on page 15, we have added a) new evidence about specific agri-food by-products derivatives which affect EGFR signaling; b) the putative mechanism of action of these natural products in relation to the EGFR modulation and c) the final biological effects resulting to be impaired. In addition, the new Figure 6 (see page 17) summarizes the paragraph.
In addition, authors need to provide a proposed mechanism of Natural flavonoids, (poly)phenol compounds in protection of tumorigenesis and cancer cell specific killing (if possible) with grouping of them.
Thanks a lot for this suggestion. Firstly, we have added a new figure (the new Figure 1 at page 2) summarizing the classification of polyphenols and the chemical structures of some representative phenolic compounds. Then, for a greater overview and impact, we have added in paragraphs 3.1, 3.2 and 3.3 the new figures (Figures 3, 4 and 5, respectively) showing the main biological effects, or the events modulated by the different natural compounds in breast, prostate and colorectal cancer, respectively.
- Although the authors described that the mechanism of action of PC and flavonoids from agri-food by products is unclear or occurs through many mechanisms, these compounds act as a radical scavenger/sink in the cell membrane, and also act as a radical buster in the active radical generation cascade, and the innate different effect between normal cells and cancer cells (activation of EGFR? or other signaling molecules) appears as a difference in sensitivity to radical/ROS. Again, the author needs to reinforce this detailed mechanism of action of these products.
Thank you, we agree. We have now described the putative mechanisms by which the different natural compounds can impair EGFR signaling. We have introduced in the new version of our review a new paragraph (Paragraph 4), titled “EGFR as putative target”. In this new section, on page 15, we have added new evidence about the putative mechanisms of action of agri-food by-product derivatives in relation to the EGFR modulation. The natural derivatives can affect the EGFR tyrosine kinase autophosphorylation and induce the EGFR downregulation in a dose- and time-dependent manner in different types of cancer cells. Again, they can inhibit the EGFR activity by disturbing the “lipid raft” and therefore impairing EGFR dimerization and activation. Furthermore, they weaken the Tyr1068 EGFR phosphorylation so affecting the receptor recycling after lipid raft alteration, inhibiting the downstream effectors ERK and Akt and the transcriptional activity of AP-1, c-fos, NF-kB, and cyclin D1 promoters in colon cancer cells. They can also inhibit the EGFR-dependent protein kinases and EGFR autophosphorylation in metastatic prostate cancer, synergistically enhancing the effect of HER2 and EGFR inhibitor, lapatinib and inhibis Akt- and ERK-signaling pathways, resulting in decreased cell proliferation in breast cancer cells. Agri-food by-products are also able to impair the cell migration and invasiveness or the EMT in different types of cancer cells through the modulation of the EGFR signaling. Thus, it is reliable that the natural products directly act on the intracellular domain of the receptor and, given their ability to influence the cell membrane plasticity (Figure 6 A in the manuscript), contribute not only to its rigidification but also to the slowing down of the receptor diffusion. This is likely to affect receptor dimerization and, in turn, its activation. Furthermore, these natural derivatives are also able to downregulate the EGFR expression through its ubiquitination and degradation (Figure 6B in the manuscript), also if the fine regulation of the degradation process is still to be determined. By these shared mechanisms, polyphenols derived from agri-food by-products are effective to impair EMT and to inhibit cell proliferation and/or migration, preferably by deactivating the signaling cascade which culminates with Akt and ERK (Figure 6C in the manuscript).
- EGFR is increased in various breast cancer subtypes, which is strongly associated with poor prognosis. The author limited the description only to IBC (minor population of BC with heterogeneity), author need to reinforce the mechanism of action for agri-food byproducts in a broader range of breast cancers with increased EGFR level.
Thank you, we have deepened the description of the breast cancer section, giving particular focus to EGFR. We have considered the importance of its expression related to the poor prognosis. We have also described the current therapies targeting EGFR and added the reference numbers of the ongoing clinical trials that are related to EGFR targeting and are unrolling patients affected by breast cancer and triple-negative breast cancer. The new discussion is in paragraph 3.1.
In addition, Line 173-174 Regarding IBC, author needs to look for more recent research articles.
Thank you. We have now added another ref. (Inflammatory breast cancer biology: the tumour microenvironment is key by Bora Lim, Wendy A. Woodward, Xiaoping Wang, James M. Reuben & Naoto T. Ueno Nature Reviews Cancer volume 18, pages 485–499 (2018) (ref 100).
- Author needs to address the clinical trial status for various diseases of various agri-food byproduct originated natural compounds.
Thank you. As mentioned in the text (paragraph 3), most of the ongoing clinical trials test the properties of the natural compounds in the context of cancer chemoprevention. We have added a table (the new table II) representing the emerging interventions based on the use of agri-food by-products in clinical trials for the management of patients affected by different types of cancer. We focused only on cancer, mainly as our review focuses on this topic.
Minor
- Quercetin has been studied widely in CRC, and the following references will be used to further enrich the content.
Rather RA, Bhagat M. Quercetin as an innovative therapeutic tool for cancer chemoprevention: Molecular mechanisms and implications in human health. Cancer Med 2020; 9: 9181-9192 [PMID: 31568659 DOI: 10.1002/cam4.1411]
Kashyap D, Mittal S, Sak K, Singhal P, Tuli HS. Molecular mechanisms of action of quercetin in cancer: recent advances. Tumour Biol 2016; 37: 12927-12939 [PMID: 27448306 DOI: 10.1007/s13277-016-5184-x]
Sharifi-Rad M, Pezzani R, Redaelli M, Zorzan M, Imran M, Ahmed Khalil A, Salehi B, Sharopov F, Cho WC, Sharifi-Rad J. Preclinical Pharmacological Activities of Epigallocatechin-3-gallate in Signaling Pathways: An Update on Cancer. Molecules 2020; 25 [PMID: 31979082 DOI: 10.3390/molecules25030467]
Thank you for your suggestion. We have extrapolated the most salient concepts about colon cancer, and we have added them in paragraph 3.3. In the same section, we have added the indicated references (160, 161, 163)
- Regarding Nobiletin and Catechin the authors need more discussion and include the following references not merely adding them.
“Nobiletin and Derivatives: Functional Compounds from Citrus Fruit Peel for Colon Cancer Chemoprevention” Cancers 2020
“Effects of Green Tea Catechins on Prostate Cancer Chemoprevention: The Role of the Gut Microbiome” Cancers 2022
Thank you we have added this evidence and the relative references in paragraphs related to colon or prostate cancer (refs 170, 146)
- Line 55 The ref 14 research was carried out in streptozotocin induced diabetic mice. This animal model is T1DM not T2DM. Author need to check it and look for the agri-food byproduct originated natural compounds on T2DM research.
Thank you, we have changed the reference to “Flavonoids and type 2 diabetes: Evidence of efficacy in clinical and animal studies and delivery strategies to enhance their therapeutic efficacy” by Hussain et al., Pharmacological Research (2020)
- Check the format of Tables and figures. Typos.
Thank you we have checked and corrected the typos.
- Table 1 Hydroxytyrosol, biophenol in the wastewater of mill seems to be Hydroxytyrosol, biophenol in olive oil wastewater of mill (OMWW)
Thank you. We have corrected the name in the new table 1
- Line 301 Ref 151 is unsuitable.
We have added all the references.
- Line 305 Needs reference
Thank you, we have added it
- There is lots of missed references in conclusion section (174, 163, 164, 165 167 175,168 etc),
We have added all the references. There was a problem with the upload in the first version. We apologize.
- Author needs to discuss the specific/common lethal mechanism of Nobiletin, naringenin, Tangeretin etc
We have specified it in paragraph 3.3
- Add the difference between catechin epicatechin and epigallocatechin
We have explained more in general what the catechins are and added the difference the first time we mention ECGC in paragraph 3.1 and added specific references.
Reviewer 4 Report
The manuscript by Sorrentino et al, seeks to highlight the potential role of food waste compounds for the treatment of cancer. There are several major concerns with the manuscript:
Firstly, the manuscript is missing about 25 references. This makes it impossible to assess the quality of the literature cited (references only go up to 153, but more are cited). While this was probably a simple mistake, it suggests the authors lack attention to detail.
Secondly, as noted by the authors themselves, these products derived from food waste target many aspects of cancer cell signalling and growth. It is unclear why they focus on EGFR. Furthermore, the evidence that targeting EGFR is a primary mechanism is weak. There needs to be a much stronger argument for the EGFR focus. The key references for EGFR are also missing (e.g. 176, 177).
There should be some discussion about the research being done to identify other compounds from food waste.
A figure showing EGFR signalling pathways and where these compounds interact would be helpful.
Minor
138: There is no reason to believe that these compounds would be safer than anti-inflammatories, indeed the examples given are quite toxic.
Author Response
Reviewer 4
The manuscript by Sorrentino et al, seeks to highlight the potential role of food waste compounds for the treatment of cancer. There are several major concerns with the manuscript:
We thank the Referee for his/her concerns and suggestions.
Firstly, the manuscript is missing about 25 references. This makes it impossible to assess the quality of the literature cited (references only go up to 153, but more are cited). While this was probably a simple mistake, it suggests the authors lack attention to detail.
Thank you very much, we noticed it. Probably, the last page relating to references has not been loaded. We have solved this unfortunate mistake and apologize very much to the reviewer.
Secondly, as noted by the authors themselves, these products derived from food waste target many aspects of cancer cell signaling and growth. It is unclear why they focus on EGFR. Furthermore, the evidence that targeting EGFR is a primary mechanism is weak. There needs to be a much stronger argument for the EGFR focus. The key references for EGFR are also missing (e.g. 176, 177).
Thank you we agree. We have now described the putative mechanisms by which the different natural compounds can impair EGFR signaling. We have introduced in the new version of our review a new paragraph (Paragraph 4), titled “EGFR as putative target”. In this new section, at page 15, we have added new evidence about the putative mechanisms of action of agri-food by-product derivatives in relation to the EGFR modulation. The natural derivatives can affect the EGFR tyrosine kinase autophosphorylation and induce the EGFR downregulation in a dose- and time-dependent manner in different types of cancer cells. Again, they can inhibit the EGFR activity by disturbing the “lipid raft” and therefore impairing EGFR dimerization and activation. Furthermore, they weaken the Tyr1068 EGFR phosphorylation so affecting the receptor recycling after lipid raft alteration, inhibiting the downstream effectors ERK and Akt and the transcriptional activity of AP-1, c-fos, NF-kB, and cyclin D1 promoters in colon cancer cells. They can also inhibit the EGFR-dependent protein kinases and EGFR autophosphorylation in metastatic prostate cancer, synergistically enhance the effect of HER2 and EGFR inhibitor, lapatinib and inhibit Akt- and ERK-signaling pathways, resulting in decreased cell proliferation in breast cancer cells. Agri-food by-products are also able to impair the cell migration and invasiveness or the EMT in different types of cancer cells through the modulation of the EGFR signaling. Thus, it is reliable that the natural products directly act on the intracellular domain of the receptor and, given their ability to influence the cell membrane plasticity (Figure 6 A in the manuscript), contribute not only to its rigidification but also to the slowing down of the receptor diffusion. This is likely to affect receptor dimerization and, in turn, its activation. Furthermore, these natural derivatives are also able to downregulate the EGFR expression through its ubiquitination and degradation (Figure 6B in the manuscript), also if the fine regulation of the degradation process is still to be determined. By these shared mechanisms, polyphenols derived from agri-food by-products are effective to impair EMT and inhibiting cell proliferation and/or migration, preferably by deactivating the signaling cascade which culminates with Akt and ERK (Figure 6C in the manuscript).
In this new section, at page 15, we have added a) new evidence about specific agri-food by-products derivatives which affect EGFR signaling; b) the putative mechanism of action of these natural products in relation to the EGFR modulation and c) the final biological effects resulting to be impaired. In addition, the new Figure 6 (see page 17) summarizes the paragraph.
We have also highlighted the importance of EGFR as a putative target in the new version of the manuscript and, in the breast cancer-related section (3.1).
There should be some discussion about the research being done to identify other compounds from food waste.
We have introduced more emphasis on this topic in the new paragraph 3. In the same paragraph, we have added a new table (table II) which summarizes the new interventions based on the use of agri-food by-product derivatives in clinical trials for the treatment of patients with different types of cancer. Additionally, in the last paragraph (Paragraph 5. Concluding remarks) we have detailed all the positive features of these molecules heightening the need for further studies that would confirm the current outcomes and may ultimately lead to new treatment options.
A figure showing EGFR signalling pathways and where these compounds interact would be helpful.
It is a good suggestion. Thus, we have added the new figure 6.
Minor
138: There is no reason to believe that these compounds would be safer than anti-inflammatories, indeed the examples given are quite toxic.
Thank you, it is an intriguing point. Thus, we took a cue from this reviewer’s clarification, and we have now introduced this concept already in the introduction section and then, more extensively, in paragraph 3. In this regard, we have also changed the title of paragraph 3 that now is “Agri-food by-products in cancer: antioxidant or cytotoxic agents?”. Thus, in our opinion, it seems more appealing. These natural compounds, indeed, can act as antioxidants or as pro-oxidants depending on their concentration. At high doses, they usually have pro-oxidant properties and act as cytotoxic agents. At low doses, the same substances exhibit opposite effects and act as antioxidants. This dichotomy of concentrations opens new challenges ad detailed in the new version of the manuscript.
Round 2
Reviewer 3 Report
The author has improved the review article according to the reviewer's opinion and considers it acceptable.
Minor
Line 59. Remove flavonoids.
Line 75 annurca apple (annurca) is correct? Malus domestica?
Line 87 author needs to put gastrointestinal tract.
Line 391-392. Author put the full name of NRF2 (nuclear factor erythroid-derived 2-like 2) two times.
Line 60. Regarding “basic polyphenol skeletons”, how about to change “aromatic hydrocarbon backbones”? (Just my opinion)
Figure 2 Plymethoxsylated should be Polymethoxslyated.
Line 397 CDK2 CDK4, whereas Line 414 CDK-2, CDK-4
Line 387 caspase-3 whereas line 414 caspase 3
There are cases where the full name and abbreviation are repeated. Author needs to correct them. (eg line281 EGF, line 414 PARP)
Carefully check typos.